# Diffusion Spectrum of Polymer Melt Measured by Varying Magnetic Field Gradient Pulse Width in PGSE NMR

**DOI:** 10.3390/molecules25245813

**Published:** 2020-12-09

**Authors:** Aleš Mohorič, Gojmir Lahajnar, Janez Stepišnik

**Affiliations:** 1Department of Physics, Faculty of Mathematics and Physics, University of Ljubljana, 1000 Ljubljana, Slovenia; janez.stepisnik@fmf.uni-lj.si; 2Institute Josef Stefan, 1000 Ljubljana, Slovenia; gojmir.lahajnar@fmf.uni-lj.si

**Keywords:** diffusion, PGSE, NMR, Rouse, reptation

## Abstract

The translational motion of polymers is a complex process and has a big impact on polymer structure and chemical reactivity. The process can be described by the segment velocity autocorrelation function or its diffusion spectrum, which exhibit several characteristic features depending on the observational time scale—from the Brownian delta function on a large time scale, to complex details in a very short range. Several stepwise, more-complex models of translational dynamics thus exist—from the Rouse regime over reptation motion to a combination of reptation and tube-Rouse motion. Accordingly, different methods of measurement are applicable, from neutron scattering for very short times to optical methods for very long times. In the intermediate regime, nuclear magnetic resonance (NMR) is applicable—for microseconds, relaxometry, and for milliseconds, diffusometry. We used a variation of the established diffusometric method of pulsed gradient spin-echo NMR to measure the diffusion spectrum of a linear polyethylene melt by varying the gradient pulse width. We were able to determine the characteristic relaxation time of the first mode of the tube-Rouse motion. This result is a deviation from a Rouse model of polymer chain displacement at the crossover from a square-root to linear time dependence, indicating a new long-term diffusion regime in which the dynamics of the tube are also described by the Rouse model.

## 1. Introduction

Molten polymers are macromolecular systems with complex translational dynamics of entangled chains and their segments being characterized by a large span of spatial and temporal scales. These dynamics are an important factor in the functionality and reactivity of the molecules. High-density entanglements, chain-bonds and cross-links prevent the formulation of an explicit theory of translational dynamics, even on larger intra-molecular length scales. In general, translation is described in a 6*N* phase space (*N* is the number of Kuhn segments of the chain) and studied by computer simulations [1,2]. Simpler models with fewer parameters can be set up. The simplest one-parameter model describing the chain motion is center-of-mass Brownian self-diffusion. In this model, the self-diffusion coefficient relates the chain center-of-mass mean square displacement (MSD) to the diffusion time: $\langle r^{2}\rangle =6D_{c}t$. Compared to the coefficients of simple liquids, this coefficient is several orders of magnitude smaller. This approach considers a polymer melt as a simple liquid and is suitable only on a long-time scale. Measurements of diffusion on a shorter time scale show anomalous diffusion [3,4,5], indicating translation more complex than that described by the Brownian model. In this case, the self-diffusion coefficient is not a constant but depends on the diffusion time and can suitably be described by its diffusion spectrum. The diffusion spectrum is the Fourier transformation of the chain segment velocity autocorrelation function.

To describe anomalous diffusion on a shorter time scale, the Rouse model [6] is used. In the Rouse model, the polymer chain is approximated by a string of Kuhn segments, connected by bonds modeled as springs, diffusing in viscous surroundings described by its effective friction drag ζ. No topological effects of the surrounding chains are considered. The MSD of a segment along a chosen axis is expressed as a sum of modes [7]:(1)$$\langle {\Delta z}_{R}^{2}\left(t\right)\rangle =2D_{c}t+\frac{4}{3}\overset{N}{\underset{p=1}{\sum }}\langle X_{p}^{2}\rangle \left(1-e^{-\frac{p^{2}t}{\tau_{R}}}\right)$$

Here, *N* is the number of Kuhn segments of length a, $D_{c}=\frac{k_{B}T}{N\zeta }$, $k_{B}$ is the Boltzmann constant, $\langle X_{p}^{2}\rangle =\frac{Na^{2}}{2\pi^{2}p^{2}}$ is the squared amplitude of displacement of the p-th mode, and $\tau_{R}=\frac{2\langle X_{1}^{2}\rangle }{3D_{c}}$ is the Rouse relaxation time of the chain. This model can be further simplified for the case of a long chain [7] to
(2)$$\langle \Delta z_{R}^{2}\left(t\right)\rangle \approx \frac{4\sqrt{\pi }}{3}\;\langle X_{1}^{2}\rangle \sqrt{\frac{t}{\tau_{R}}}$$

Intermolecular entanglements in a dense polymer prevent the lateral motion of a chain and localize it inside a curved tube. The Rouse model can be used to model a shorter part of the chain, with $N_{e}$ Kuhn segments, between the adjacent entanglements in the short time limit. In the intermediate time regime, segments reach the tube walls, and their motion is restrained. The polymer chain can only move along the tube in a reptation process [8,9]. As the polymer chain is released from the tube, the correlation with the initial conformation is lost. The described progression in translational mechanisms causes successively different time dependencies of the MSD: proportional to $t^{1/2}$ at short times, to $t^{1/4}$ and back to $t^{1/2}$ in the intermediate reptation regime, and to t for the chain disengagement [10]. Considering the collective motion of chains as a single chain moving in a fixed tube is a simplification. In real polymers, adjacent chains also move, causing constraint release [11,12], and the tube itself behaves as a coarser-grained Rouse chain [8,13], with the relaxation time proportional to the lifetime of the obstacles. The combined model of chain reptation inside the tube, exhibiting Rouse motion, predicts the segmental MSD starting as $t^{1/2}$ and evolving to a t time dependency. According to [11], the longest relaxation time for the tube-Rouse motion τ in a mono-dispersed polymer melt is equal to the terminal time of the chain’s reptation in the tube, which amounts to almost equal contributions of both processes to the MSD $t^{1/2}$ dependence.

An alternative to the description of the MSD in the time domain is the transformation of the translation dynamics to the frequency domain, where it is described by the power spectrum D(ω)=−ω^2/2 FT[〈Δz^2(t)〉], where FT is the Fourier transformation. As the MSD exhibits different translational modes expressed through changes in the power of the MSD time progression, so does D(ω). It starts as a constant $D_{c}$ at low frequencies and increases as $\omega^{1/2}$ and $\omega^{3/4}$ in the tube/reptation regime. It passes into the Rouse regime at the frequency $2\pi /\tau_{R}$ and exhibits $\omega^{1/2}$ dependence again until it levels off at high frequencies [14,15]. The tube/reptation model replaces the many-chain problem by a single chain moving in a tube of topological constraints exerted by the surrounding chains. This model oversimplifies the actual dynamics, because the surrounding chains are not static but are moving as well. This motion is responsible for the constraint release and relaxation of the tube [11,12]. The tube relaxation time for constraint release, or tube reorganization, is determined by obstacle lifetime [8]. Various models account for the impermanence of entanglements [16,17,18]. The theory of constraint release involves tube dilation and tube-Rouse motion [13]. In this model, the constraint release is considered as a Rouse motion of the tube with coarser segments and slower relaxation than the Rouse motion of the chain. The chain relaxation in polymer melts results from two independent and concurrent processes: reptation inside the tube and tube-Rouse motion as the tube reorganization. The diffusion spectrum at low frequencies starts from a constant for both processes and changes to the $\omega^{1/2}$ at inverse values of the longest relaxation time of each process. In the case of a mono-dispersed polymer melt, the tube reorganization is slower than reptation and must have a small effect on the diffusion properties [16,17]; nevertheless, tube reorganization significantly affects the viscoelastic properties of the polymer melt, presumably because of the difference between the spectrum of the tube-Rouse modes and the spectrum of reptation [11].

The structural and dynamical properties of polymers predicted by the models are in qualitative agreement with experimental data resulting from different methods, some of which are not limited to a macroscopic, rheological scale and offer insight into the chain dynamics on the segmental scale [12]. In a polymer melt, the chains exhibit a complex hierarchy of dynamic processes. Very fast and local conformational rearrangements on the picosecond scale can be measured by neutron scattering [19]. Slow, diffusive and cooperative motion extending into the range of seconds can be observed by methods of nuclear magnetic resonance (NMR), optical methods or viscosity measurements in rheology [20]. NMR is sensitive to polymer dynamics on a wide range of time scales; for example, the diffusion coefficient can be measured in the interval from milliseconds to seconds [3], either indirectly with NMR relaxometry [10,21,22,23] or directly by measuring the effect of spin-bearing-molecule displacement on the gradient spin-echo (GSE) attenuation in the applied magnetic field gradient [3,10]. 

The chain translation dynamics influence NMR relaxation because the dipolar coupling between adjacent spins depends on mutual orientation. The orientational fluctuations mirror the segmental dynamics through the magnetic dipole–dipole correlation function [24,25]. The correlation function includes intramolecular and intermolecular contributions. Intramolecular interactions fluctuate due to molecular rotation. Intermolecular couplings also depend on the relative translational motion of the chains. The presence of internal field gradients (conditioned by voids in polymer melts) in high-molecular-mass polymers has been suggested in [10] based on an accelerated transversal relaxation rate obtained from free induction decay, which is effectively reduced by the application of a Carr–Purcell–Meeboom–Gill pulse sequence. Other phenomena can lead to accelerated relaxation, e.g., dipole–dipole interactions not averaged by molecular motion arising due to the anisotropy of the motion of the chain segments, which is typical for entangled polymer chains. Reorientational and translational dynamics must be discerned in order to study polymer dynamics by NMR relaxometry. This is achieved by different techniques, e.g., by the isotope dilution technique in field-cycling and transverse NMR relaxometry [10,26], by combining NMR relaxometry and dielectric spectroscopy [27,28], or by double-quantum NMR experiments [22,23,29]. Different models and approximations of polymer dynamics have been discussed in the context of different spin relaxation studies, failing to provide an exact form of the correlation function; however, these experiments generally confirm the scaling laws of the reptation model [25]. 

GSE methods can be roughly divided into two classes, modulated and pulsed GSE. Modulated GSE (MGSE) uses an applied magnetic field gradient modulated in a way to harmonically change spin dephasing, thus measuring the diffusion spectrum at the modulation frequency [30]. Pulsed GSE (PGSE) employs (two) short gradient pulses separated by a defined time interval. In the limit of short pulses, this time interval can be considered as the diffusion time. The first applied gradient pulse defocuses spins, encoding their position in their phase; the second, decoding pulse refocuses all stationary spins in the spin echo. The moving spins do not refocus completely, causing the attenuation of the echo, which thus becomes sensitive to translational motion. If the time between the pulses, the diffusion time, is longer than the terminal relaxation time, defined as the asymptotic viscous decay of the polymer in rheology, the PGSE method can provide the polymer center-of mass diffusion coefficient in the polymer melt [4]. PGSE can also measure anomalous diffusion. The shortest diffusion time interval is limited by the strongest applicable gradients, and the longest diffusion time interval is limited by the decoherence of spins (transversal relaxation). This puts the limits of the segment displacement that can be detected by GSE NMR somewhere in the range of several hundred nanometers, assuming the self-diffusion coefficient of the high-molecular-weight polymer melt is on the order of 10^−15^–10^−12^ m^2^·s^−1^. Polymer chain reptation displacements are smaller than 100 nm and are not detectable with a conventional PGSE experiment. Conflicting reports on the self-diffusion *N* scaling power follow from poorly determining the center-of-mass diffusion coefficient without considering the crossover to the anomalous diffusion regime at the same time [3]. 

Internal gradients, caused by susceptibility mismatches or paramagnetic centers, can cause artifacts, leading to the overestimation of the self-diffusion coefficient. There are common NMR diffusion techniques that can be used to reduce artifacts by internal gradients, such as bipolar gradients [31]. However, when the background gradients are spatially non-uniform, molecular diffusion introduces a temporal modulation of the background gradients, defeating the simple bipolar gradient suppression of background gradients in diffusion-related measurements. Several other methods have thus been proposed to minimize the effect of the internal gradient [32,33,34,35], among which is also the method presented in the paper [14], where the data from the PGSE measurements are explained with a crossover to the anomalous diffusion regime in polymer melts with the addition of the internal gradient effect. In certain cases, the effect of internal gradients can provide valuable information on the dynamics, topology or composition of the material studied [36]. Measurements of molten polydisperse polymers provide a diffusion coefficient that scales as $N^{2}$ for polymers with numbers of Kuhn segments larger than the entanglement number and as power N for those below [4]. However, the subsequent PGSE measurements of very mono-dispersed molten polymer [37,38] do not confirm this result but provide a scaling power larger than 2 for the total range of polymer lengths without any crossover to the power 1 for short chains. These conflicting data could result from a mis-defined crossover to the anomalous diffusion regime as shown in [30]. There are also reports that crossover is mis-defined because a strong internal susceptibility magnetic field at the interstices of voids in a polymer melt spoils the measurement [12,15]. Internal gradients are, aside from paramagnetic centers, caused by voids in the melt. Voids in polymer melts are statistically varying formations, which can be characterized by their sizes and mean lifetimes. For example, in polybutadiene, these voids are adjacent to the reptating chain segments and characterized by a diameter ~0.5 nm. However, such voids can affect the diffusion NMR experiment only if the diameters of the voids and mean lifetimes are at least of the order of magnitude of the covered diffusion paths and the diffusion times, respectively [39]. A method that also avoids the effects of the internal gradient is the MGSE method. Its results for self-diffusion measurements of mono-dispersed molten polymers [37,38] provide scales for the total range of polymer lengths and the transition into the regime of entanglement at Kuhn steps, which are below theoretical predictions. A test of the tube/reptation model by measuring the diffusion of nanoscopic strands of linear, mono-disperse poly (ethylene oxide) embedded in artificial cross-linked methacrylate matrices is described in [40]. PGSE studies of polymer dynamics are well described by the Rouse model in the case of dilute and semi-dilute polymers [41,42,43,44]. However, the PGSE measurements of diffusion in dense polymers do not clearly support the tube/reptation model [5,10,37,45]. The MGSE method, which measures the velocity autocorrelation spectrum, shows that in a polymer melt, the tube-Rouse motion has a prevailing role at long diffusion times, and this indicates faster tube reorganization than expected [30].

NMR measurements in a magnetic gradient field are sensitive to the MSD, $\langle \Delta z^{2}\rangle ,$ in the direction of the applied magnetic field gradient G=∇|B|, here taken to be along the z axis. The attenuation of the spin echo is given by:(3)$$\ln\frac{S_{0}}{S}=\frac{1}{\pi }\int_{0}^{\infty }{\left|q\left(\omega \right)\right|}^{2}D\left(\omega \right)d\omega$$
where S is the spin-echo amplitude, $S_{0}$ is the amplitude of the echo without the applied gradient (in the limit G→0) and ${\left|q\left(\omega \right)\right|}^{2}$ is the sampling function tailored by the gradient–radiofrequency pulse sequence. The gradient sampling function for the Hahn-echo PGSE sequence is given by [34]:(4)$${\left|q\left(\omega \right)\right|}^{2}=16\gamma^{2}G^{2}\frac{\sin^{2}\frac{\omega \delta }{2}\sin^{2}\frac{\omega \Delta }{2}}{\omega^{4}}$$

Here, G is the strength of the gradient, γ is the gyro-magnetic ratio, δ is the width of the gradient pulse and Δ is the time interval between the leading edges of the two gradient pulses. The gradient sampling function Equation (4) is shown superimposed on the diffusion spectrum given by Equation (5) in Figure 1.

This paper presents a study of anomalous self-diffusion in a linear polyethylene melt by the PGSE method. A special short diffusion time sensitivity is achieved by the variation of the gradient pulse width δ, contrary to usual measurements of anomalous diffusion with variable inter-pulse separation Δ. By changing only δ, artifacts induced by internal gradients can also be reduced as described by Equation (A2) in [14]. Measurements with PGSE are more effective for long diffusion times or, conversely, the low-frequency part of the diffusion spectrum. A problem arises if we want to measure the diffusion spectrum at high frequencies, as a short Δ together with strong magnetic gradients must be used to achieve the desired attenuation of the spin echo. This is experimentally hard to implement. Additional attenuation caused by the background or internal magnetic field gradient and the effect of transverse relaxation must also be accounted for when the inter-pulse separation is changed. Here, we set out to verify the results of the measurements of a polymer melt diffusion spectrum with the MGSE method reported in [15] by an alternative method of PGSE. The results in [15] show that the observed dynamics in the low-frequency range belong to tube-Rouse motion [13] and can be described by the formula
(5)$$D\left(\omega \right)=D_{c}+D_{s}\frac{\omega^{2}\tau^{2}}{1+\omega^{2}\tau^{2}}$$
where $D_{c}$ is the center-of-mass diffusion coefficient, $D_{s}$ is the diffusion rate of the tube segments and τ is the tube-Rouse time, corresponding to the characteristic time of the crossover. This spectrum is shown in Figure 1 and overlaid with the gradient sampling function of the PGSE sequence.

For the diffusion spectrum model given in Equation (5), the spin-echo attenuation Equation (3) becomes:(6)$$\ln\frac{S_{0}}{S}=\gamma^{2}G^{2}\delta^{2}\left(\Delta -\frac{\delta }{3}\right)D_{c}+\gamma^{2}G^{2}\tau^{3}D_{s}\left[2e^{\frac{\Delta }{\tau }}\left\{1+\left(\frac{\delta }{\tau }-1\right)e^{\frac{\delta }{\tau }}\right\}-{\left(e^{\frac{\delta }{\tau }}-1\right)}^{2}\right]e^{-\frac{\delta +\Delta }{\tau }}.$$

The standard evaluation of the diffusion data measured with the PGSE is calculating the effective diffusion coefficient $D_{e}$, defined with:(7)$$\ln\frac{S_{0}}{S}=-bD_{e},$$
where the b factor is given by $b=\gamma^{2}G^{2}\delta^{2}\left(\Delta -\frac{\delta }{3}\right)$. The effective diffusion coefficient is a constant for all possible parameters in the case of Brownian diffusion, but in the case of anomalous diffusion, it is interpreted as a time-dependent diffusion coefficient, in our case, given by:(8)$$D_{e}=D_{c}+\frac{\tau^{3}}{\delta^{2}\left(\Delta -\frac{\delta }{3}\right)}D_{s}\left[2e^{\frac{\Delta }{\tau }}\left\{1+\left(\frac{\delta }{\tau }-1\right)e^{\frac{\delta }{\tau }}\right\}-{\left(e^{\frac{\delta }{\tau }}-1\right)}^{2}\right]e^{-\frac{\delta +\Delta }{\tau }}.$$

## 2. Results and Discussion

Polyethylene melt diffusion was measured by the PGSE method. In the experiment, the spin-echo amplitude was recorded by varying the gradient pulse width δ at several different strengths of the applied gradient pulse. Figure 2a shows the spin-echo amplitude and (b) the derived effective self-diffusion coefficient as defined in Equation (7), both as a function of the gradient pulse width δ and for all the applied gradient strengths. The effective diffusion coefficient in Figure 2b clearly shows signs of anomalous diffusion.

The model describing the data is given by Equation (8), and a few simplifications can be made, since it is reasonable to assume from previous measurements [30] that Δ≫τ, to obtain a simpler model:(9)$$D_{e}=D_{c}+\frac{2\tau^{3}}{\delta^{2}\left(\Delta -\frac{\delta }{3}\right)}D_{s}\left[e^{-\frac{\delta }{\tau }}+\left(\frac{\delta }{\tau }-1\right)\right].$$

Both models return the same fitting parameters with the measured data. A least-square non-linear fit of the model to the data gives the parameters presented in Table 1.

The only value estimated with high certainty is the chain diffusion coefficient $D_{c}$. Both the tube segment diffusion coefficient and the relaxation time appear in the model (Equation (9)) together as $\tau^{2}D_{s}$ to the first order of τ/δ, and any change in one can be compensated by an according change in the other and does not significantly alter the fit. Thus, the δ used in the measurements should be accordingly short, or at least one of the parameters should be determined separately. The results for the chain diffusion coefficient match within the error with the results in [14] and [46]. The result for the relaxation time matches the tube displacement per obstacle lifetime $L_{eq}/\sqrt{\tau_{ob}}$, if the number of Kuhn segments between the entanglements $N_{e}$ is 25 (compared to the 120 total number of segments per chain) since $L_{eq}=\frac{N}{N_{e}}a$ and $\tau =\frac{N^{2}}{N_{e}^{2}}\tau_{ob}$.

The dashed line in Figure 2b represents the best fit of the $D_{e}$ model to the data without the points measured at δ=1 ms. The fitting parameters in this case differ significantly: $D_{c}=3.3\times 10^{-13}\;m^{2}/s$, $D_{s}=1.8\times 10^{-11}\;m^{2}/s$ and τ=3.8 s. This demonstrates the sensitivity of the model to the input data without measurements at a short-enough δ, which should be short enough to include the increase in $D_{e}$ at a short δ. It also demonstrates that caution using the approximation Δ≫τ for the model in Equation (9) is warranted, since τ was determined to be longer than Δ in the case of short δ data points being excluded. 

We have shown here that the PGSE method enables the measurement of the segmental translation of polymeric chains by the variation of the gradient pulse width. This approach can also effectively take into account the effect caused by the internal gradient, which commonly affects PGSE measurements but requires knowledge of the interplay between the molecular motion and the buildup of spin phase structure during the magnetic field gradient action. By combining the PGSE sampling function and the segmental diffusion spectrum rendered from the model of tube-Rouse motion [13], we obtain the dependence of the PGSE signal attenuation on the gradient pulse width. The data obtained by the measurements of molten polyethylene well fit the predictions and provide evidence of the tube-Rouse motion model proposed in [13]. The model, which was already confirmed for other polymer samples by the MGSE method [30], reveals the tube segmental motion in the range of milliseconds. The sample polymer data $M_{w}$ and $M_{n}$ indicate a sharp distribution of fragment sizes; thus, the effects of polydispersity, which may cause a deviation from the model, can be neglected in our case.

This study presented here is a reevaluation of the study in [14]. The study of polymer diffusion by the MGSE method [30] shows slow dynamics that can be attributed to the reorganization of the polymer tube with temporal and spectral resemblance to Rouse motion. This description matches the theory of tube-Rouse motion put forward in [13]. In [14], PGSE measurements were used to trace the crossover of the chain Rouse dynamics from $\sqrt{t}$ to t dependence because of the constraint release. The constraint release was originally termed tube reorganization by Pierre-Gilles de Gennes, where the obstacle lifetime determines the tube relaxation times [8]. Various models account for the impermanence of entanglements [16,17,18], among which is also the theory of constraint release involving the tube dilation and tube-Rouse motion [13]. In this theory, the constraint release is considered as the tube-Rouse motion, and the relaxation time is proportional to the lifetime of the obstacles [13]. In the previous paper [14], we followed a quite common approach to considering polymer chain dynamics described by the Rouse model in the range where the dynamics cross from square-root to linear time dependence in the MSD [47], to explain the anomalous effective diffusion obtained from the PGSE measurements in polymers. In the original experiment, the internal gradient artifacts were not suppressed by any of the numerous methods, because the system was considered homogenous enough and the measurements fitted well to the model used for the larger part of the measured interval. However, the results deviate from the model in the limit of the short δ. In [14], it was proposed that the deviation was a result of internal gradients (caused by a susceptibility mismatch) adding to the effect of the external applied gradient. An extra term based on internal gradients was added to the attenuation factor, resulting in a better fit. According to [39], this would require unrealistic conditions, and a search for a better explanation was fruitful, since the results are here satisfactorily described with the new tube-Rouse model and without recourse to the effect of the internal gradient. This is also in accordance with subsequent measurements with the MGSE method [30], which indicate that the polymers in the millisecond time range exhibit some new dynamics that are not related to the motion of the polymer chain inside the tube, but can be explained by the theory of polymer tube reorganization, where the tube behaves in a similar way to a chain, and therefore, this motion can be called tube-Rouse motion [13]. In this paper, we show that the new interpretation fits better to the results of our PGSE measurements. We show that the data can be well fitted to this model of tube dynamics, and this is a deviation from the previous results based on long-chain approximation (Equations (4) and (6) in [14]). 

## 3. Materials and Methods 

We studied a sample of linear polyethylene Standard Reference Material 1482 with a narrow molecular weight distribution ($M_{n}=11,400\;{g\;\mathrm{mol}}^{-1}$, $M_{w}=13,600\;{g\;\mathrm{mol}}^{-1}$) prepared by NIST, Washington, DC, USA. Measurements were performed on a melted polyethylene sample at 426 K.

The measurements were performed on a home-made pulsed NMR spectrometer (Ljubljana, Slovenia) at a 60 MHz proton NMR frequency and equipped with a magnetic field gradient coil system described in [48]. The PGSE sequence is shown in Figure 3. The widths of the π/2 radiofrequency (RF) pulses used were 1.2 microseconds. The π RF pulse was applied symmetrically between the gradient pulses. The gradient pulse followed the RF pulse with a delay short enough to be neglected in the signal analysis. The same is true for the echo following the second gradient pulse; however, the echo followed the second gradient pulse with a delay large enough that no artifacts were introduced because of the finite gradient fall time. The PGSE attenuation dependence on the duration of the gradient pulses was measured by changing the pulse width δ from 1 to 15 ms, with the diffusion time (the interval between the gradient pulses) fixed at Δ=80 ms. The measurements were performed with the gradient fields 4.38, 3.04 and 1.34 T/m.

## Figures and Tables

**Figure 1 molecules-25-05813-f001:**
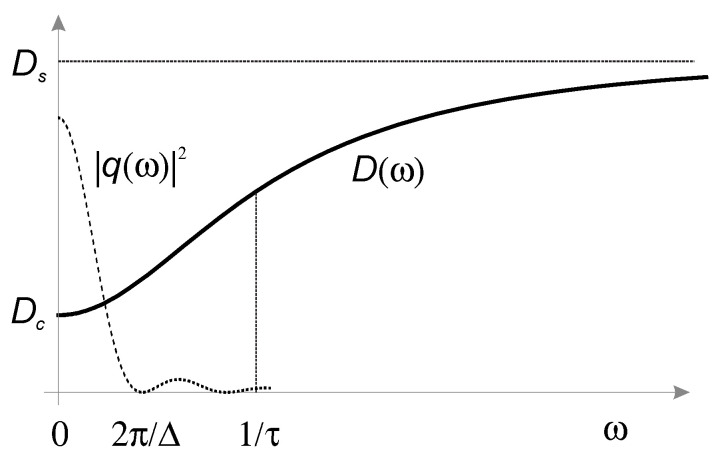
The diffusion spectrum D(ω) of a polymer melt and the sampling function ${\left|q\left(\omega \right)\right|}^{2}$ of the pulsed gradient spin-echo (PGSE) sequence.

**Figure 2 molecules-25-05813-f002:**
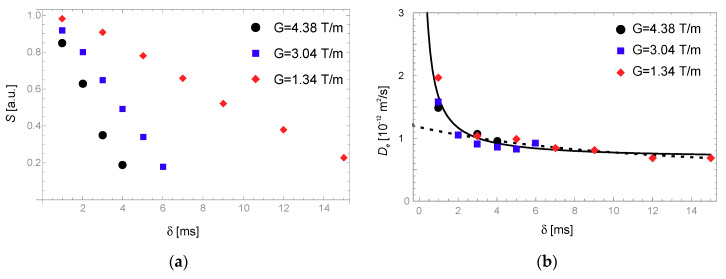
Measurements of polyethylene melt self-diffusion with the PGSE and a fixed diffusion time Δ=80 ms at temperature 426 K: (**a**) The echo amplitude as a function of the applied gradient pulse width for different applied gradient strengths; (**b**) The effective diffusion coefficient as a function of the applied gradient pulse width. Included are the least-square fits of the model in Equation (9) with the best-fit parameters for all the data (solid line) and excluding the shortest δ points (dashed line).

**Figure 3 molecules-25-05813-f003:**
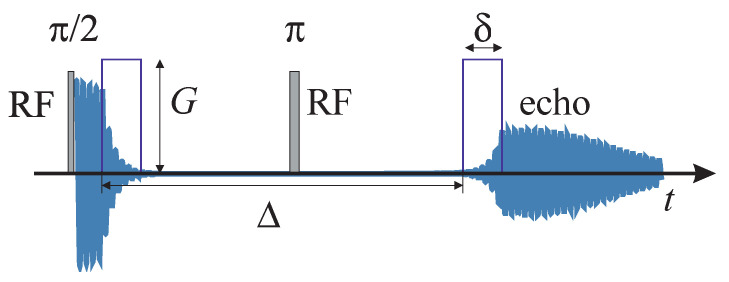
The PGSE sequence used to measure self-diffusion—shown are the RF and gradient pulses and the NMR signal: free induction decay at short times and echo after time Δ (blue line).

**Table 1 molecules-25-05813-t001:** Parameters of the effective diffusion model of Equation (9) of the polyethylene melt SRM 1482 at 426 K.

Parameter	Estimate	Standard Error	t-Statistic	*p*-Value
$D_{c}$	7.0 × 10^−13^ m^2^/s	8.0 × 10^−14^ m^2^/s	9.3	2.2 × 10^−7^
$D_{s}$	5.8 × 10^−11^ m^2^/s	7.0 × 10^−10^ m^2^/s	0.51	0.62
τ	0.75 ms	0.60 ms	1.25	0.23

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
