# Peer review of "Diffusion Spectrum of Polymer Melt Measured by Varying Magnetic Field Gradient Pulse Width in PGSE NMR"

_molecules, 2020, doi:10.3390/molecules25245813_

Round 1

Reviewer 1 Report

  1. In the PFSGU NMR method, the amplitude of the field gradient usually changes, but not the inter-pulse separation, as the authors wrote.
  2. In the used version of PGSE, authors vary gradient pulse width at a fixed diffusion time and quite high amplitude of the pulsed field gradient. They loose contributions of the fast-moving components of the system. There is no any advantages of such PGSE NMR experimental approach.
  3. Authors mentioned that their approach allows to reduce artifacts caused by internal magnetic field gradients. However, it is not clear what is the source of these gradients in a homogeneous polymer melt. Probably, this has no sense.

Reviewer 2 Report

See attached review

Reviewer 3 Report

In their current paper, the authors present their old diffusion data acquired on polyethylene polymer melt. The original paper entitled “Study of translational dynamics in molten polymer by variation of gradient pulse” is referred to as the Reference 14, published in Journal of Magnetic Resonance, 2013. The difference between the two papers is in the equation used to fit their data (compare Fig. 7 of Ref. 14 and Fig. 1b of this paper). However, in the present paper, there is no discussion about the previous work and results are not compared. It is not clear what are new results or new conclusions. This is my major objection and I suggest substantial revisions. Besides that, I highlight other issues below.

The last sentence in Abstract states determination of “the characteristic relaxation time of the first mode” and “establishing “the deviation from the model”.  The “time” is introduced in Eq. 3 as “the characteristic time of the cross-over” which I find confusing. Moreover, the authors admit strong correlation between the “time” and “the diffusion rate of tube segment” which makes determination of these parameters impossible. I have not found in the text any discussion of “deviation of the model”. The abstract does not seem right.

Material and Methods section contains a lot of text (lines 130-201) better suited for Introduction or Theory sections. The effective diffusion constant is determined from the ratio of signal intensities, but nothing is said about what the reference signal is and how it is obtained.

Results section begins with statement that the effective diffusion constant was obtained from Eq. 9 but in fact Eq. 4 was used. Figure 1b presents two fitted lines but only one set of parameters is given in Table 1 – these are the result of which fit? The two fits are not discussed at all.

In the text, below line 154, it is discussed that Modulated GSE is preferred over Pulsed GSE. I find it confusing then that Modulated GSE is in fact experimentally realized as Pulsed GSE, see the line 186.

Author Response

The authors would like to thank the reviewer for great remarks and a big help to improve the paper.

Reviewer 4 Report

The manuscript "Diffusion spectrum of polymer melt measured by varying magnetic field gradient pulse width PGSE NMR" repot an interesting piece of information about diffusion in polymer melts.

The paper is overall well written even if the organisation of the sections need improvements in my opinion. In particular, I suggest to insert a "Theory" section where the model used is described and to rearrange the equation's order ( for example Eq.4 is derived from eq. 9 but is reported much earlier, this is confusing in my opinion). Also, lines form 180 to 199, now in "Materials and Methods" section should be move into a theory section. Moreover, the first part of "Materials and Methods" section (lines 130 to 179) should be moved to introduction. I also suggest to join the results and discussion part.

Another important issue to be addressed by the authors, is the following: in ref. 14 [Journal of Magnetic Resonance 236 (2013) 41–46, by the same authors] Deff values for the same sample are shown. Those results are a bit different with respect to those reported in the present paper. Please comment on this point and better clarify the difference between the two data sets and/or analysis.

Other minor points are:

line 106 caption of figure1. It is not clear what is the dashed line.
line 204 please specify the pulse durations.

Author Response

(The authors gave the same response as above.)

Round 2

Reviewer 1 Report

  1. The presence of internal field gradients (conditioned by voids in polymer melts) in high molecular mass polymers has been suggested by Kimmich et al. (Kimmich and Fatkullin, Adv. Polym. Sci. 2004, 170, 1-113) based on accelerated relaxation T2 rate obtained from free induction decay, which is effectively reduced by the application of CPMG pulse sequence. Mohoric et al., authors of the current paper, took this suggestion without any criticism. Any other phenomena, which can lead to accelerated relaxation, did not take into account. For example, non-averaged by molecular motion dipole-dipole interaction arising due to anisotropy of motion of the chain segments, which is typical for entangled polymer chains, is the most evident reason for this. There are experimental NMR relaxation methods, which allows distinguishing effects of background (internal) gradients and effects of the anisotropic motion of polymer chains.
  2. There are common NMR diffusion techniques, which can be used to reduce artifacts by internal gradients, such as bipolar gradients (Cotts et al., J. Magn. Reson. 1989, 83, 252). Why authors did not apply such sequences.
  3. Concerning voids in polymer melts and its effects on internal field gradients. These are statistically varying formations, which can be characterized by size and mean lifetime. For example, in polybutadiene, these voids are adjacent to the reptating chain segments and characterized by diameter ~0.5 nm. However, according to Kärger et al., (Kärger, Pfeifer and Henk, Adv. Magn. Reson.: V. 12, P. 67.) such voids can affect the diffusion NMR experiment only if the diameter of voids and mean lifetimes are at least of the order of magnitude of the covered diffusion paths and the diffusion times, respectively. In the manuscript of Mohoric et al. molecular displacements estimated at the shortest diffusion time is 200-450 nm. Therefore, the authors suggest the presence of such large voids with a lifetime longer than 80 ms (diffusion time of the experiment). If such large and long-living voids present in the melt, what is inside the voids? Vacuum? Air?
  4. The quality of the experimental data is poor. The range of diffusion decays in Fig.2a is less than one decimal order, with only a few points shown in the decay. The presentation of decay is very strange. There is a common presentation as lg(S) – (delta)2. From the linear presentation of the decay is nothing (such as anomalous diffusion) evident.

Reviewer 2 Report

The authors have carefully responded to my previous questions and concerns. The clarification of the difference with their previous report has been strengthened.

In my opinion the paper is now acceptable for publication.

Nice job.

Author Response

The authors would like to thank the reviewer for helpful comments that improved the quality of the manuscript.

Reviewer 3 Report

The authors addressed most of the points adequately. I only regret the study contains poor experimental data.

Author Response

(The authors gave the same response as above.)

Round 3

Reviewer 1 Report

The manuscript was corrected and can be accepted for publication.